# Tenofovir alafenamide is superior to tenofovir disoproxil fumarate and entecavir in cost-effectiveness of treatment of chronic hepatitis B in china with new volume-based procurement policy

Yi Lin[1,2,3], Xueyan Lin[1,2,3], Ruiqi Xia[1,2,3], Juan Chen[1,2,3], Zhihui Lin[1,2,3]*, Shiyun Lu [1,2,3]*

1 Shengli Clinical Medical College of Fujian Medical University, Fuzhou, China, 2 Gastroenterology and Hepatology Department, Fujian Provincial Hospital, Fuzhou, China, 3 Department of Gastroenterology and Hepatology, Fuzhou University Affiliated Provincial Hospital, Fuzhou, China

These authors contribute equally

* 13799371368@139.com (ZL); lushiyun121739@163.com (SL)

## Abstract

### Background

Evidence supports the long-term efficacy of Nucleos(t)ide Analogs (NAs) therapy in improving chronic hepatitis B (CHB) prognosis. However, determining the most cost-effective first-line NAs remains unclear. China's implementation of the New Volume-Based Procurement Policy (NVBP Policy) in 2019 led to substantial price reductions for entecavir (ETV), tenofovir disoproxil fumarate (TDF), and tenofovir alafenamide (TAF). This study assesses the cost-effectiveness of ETV, TDF, and TAF, both with and without NVBP, for CHB in China.

### Methods

A state-transition model, parameterized using data from published literature, was utilized to compare treatment strategies encompassing non-NAs best support care (BSC), ETV, TDF, and TAF, with or without NVBP. A simulated lifetime cohort was employed, measuring outcomes such as predicted liver-related deaths, costs, quality-adjusted life-years (QALYs), and incremental cost-effectiveness ratios (ICERs).

### Results

In comparison to Non-NAs BSC, TAF yielded an additional 2.68 QALYs per person, with an ICER of 7,853.22 USD/QALY. Subsequently, TDF generated an additional 2.61 QALYs/person at an ICER of 7,153.39 USD/QALY, and ETV produced an additional 2.01 QALYs/person with an ICER of 9,366.74 USD/QALY without NVBP. Incorporating NVBP, the ICERs for TAF, TDF, and ETV decreased to −745.62 USD/

**Data availability statement:** All relevant data are within the paper.

**Funding:** This study is supported by the Fujian Provincial Natural Science Foundation, Grant No.2020J05264 & 2023J011210; Science and Technology Planning Project of Fujian Provincial Health Commission, Grant No. 2020GGA002; Sponsored by Fujian provincial health technology project, Grant No. 2022QNA003.The funding source had no involvement in the study design; in the collection, analysis, or interpretation of data; in the writing of the manuscript; or in the decision to submit the article for publication. The authors declare no financial or non-financial conflicts of interest related to this research.

**Competing interests:** The authors have declared that no competing interests exist.

QALY, −729.33 USD/QALY, and −871.11 USD/QALY, respectively, compared to non-NAs BSC. At willingness-to-pay (WTP) thresholds ranging from 12,500 USD/QALY to 37,500 USD/QALY, TAF with NVBP showed an increased probability (51.15–52.47%) of being the optimal treatment strategy, followed by TDF and ETV with NVBP exhibiting a reduced likelihood 43.09–42.45% and 6.40–4.48% in the iterations.

## Conclusions

Our analysis suggests that TAF with NVBP represents the most cost-effective long-term therapy for CHB. Both TDF and ETV, with or without NVBP, and TAF without NVBP were considered cost-ineffective.

---

## Introduction

Chronic Hepatitis B (CHB) constitutes a significant global health burden, affecting an approximately 292 million individuals worldwide [1]. In regions of high prevalence, such as China, the prevalence of the hepatitis B surface antigen (HBsAg) stood at approximately 6.89% (95% CI: 5.84–7.95%) in 2018, reflecting an estimated populace of 84 million living with HBsAg [2]. Notably, due to this high HBsAg prevalence, China accounts for 11% of global cirrhosis deaths [3] and more than 50% of the global burden of liver cancer [4].

Antiviral treatment represents the recommended approach to mitigate cirrhosis and hepatocellular carcinoma (HCC) incidence in CHB patients, as endorsed by guidelines [5–7]. Specifically, Entecavir (ETV), tenofovir disoproxil fumarate (TDF), and Tenofovir Alafenamide (TAF) are nucleos(t)ide analogs (NAs) with high resistance barriers, recommended as first-line therapy by the American Association for the Study of Liver Diseases (AASLD), the World Health Organization (WHO), and the Chinese Society of Hepatology (CSH) [5, 7–9]. While these NAs effectively suppress HBV DNA replication, they do not eliminate covalently closed circular DNA (cccDNA) or viral DNA integrated into the host genome [10]. Notably, HBV viremia tends to recur upon treatment cessation, even after successful virus suppression during therapy [11]. Consequently, the widely accepted criteria for terminating therapy duration remain unclear.

Recent evidence underscores the ability of both long-term ETV and TDF therapies to reduce cirrhosis, HCC incidence, and CHB mortality [12–16]. Moreover, some studies suggest that TDF may exhibit a greater preventive effect against cirrhosis and HCC than ETV [17–20]. However, concerns regarding bone, phosphorus, and renal function risks associated with TDF have been reported [21]. Both Chinese chronic hepatitis B guidelines and Asian-Pacific clinical practice guidelines recommend regular surveillance of bone mineral density, blood phosphorus levels, and renal function [7,22]. Frequent surveillance every 3–6 months might elevate the total cost of lifelong therapy [22]. Considering that China's average healthcare expenditure remains lower than the global average [23], analyzing which NA presents an affordable and highly cost-effective long-term therapy strategy becomes imperative.

Prior studies have discussed the cost-effectiveness of ETV or TDF in China [24–26]. However, the introduction of TAF treatment in late 2018, offering efficacy similar to TDF but boasting superior renal and bone safety profiles, has not yet been extensively explored [21,27, 28]. Another crucial factor is the implementation of China's New Volume-Based Procurement Policy (NVBP) in public hospitals nationwide from 2019. Under this policy, the Medical Insurance Department organized a public hospital procurement alliance to negotiate with pharmaceutical manufacturers, aiming to lower drug prices while ensuring quality medicine accessibility via large-scale procurement, known as volume-based pricing [29]. Both branded and generic drugs were eligible for tender, with generic drugs mandated to pass the State Drug Administration's evaluation for consistency in quality and efficacy before bidding [29,30]. By 2023, prices of ETV, TDF, and TAF under new volume-based procurement (NVBP) had been reduced by 99.2%, 98.7%, and 96.3%, respectively [31,32]. Some international studies indicate that the cost-effectiveness of TAF in high-income countries is constrained by its high pricing. For instance, Tian et al. found in Canada that TAF would need a 33.4% price reduction to achieve comparable cost-effectiveness with TDF and ETV [33]. Similarly, Buti et al. (2021) reported in their study in the European Union that the incremental cost-effectiveness ratio (ICER) for TAF generally exceeds 50,000 USD/QALY, significantly surpassing local willingness-to-pay thresholds [34]. Against this backdrop, little is known about the cost-effectiveness of first-line NAs for CHB treatment in China

This study seeks to model and analyze the cost-effectiveness of ETV, TDF, and TAF, with or without NVBP, in the context of long-term CHB therapy. Our findings provide direct evidence for Chinese healthcare policymakers to prioritize TAF procurement under the NVBP framework, thereby optimizing medical resource allocation. Concurrently, clinicians can utilize these cost-effectiveness findings to promote TAF adoption in resource-limited settings. Furthermore, the success of the NVBP offers a potential model for other low- and middle-income countries to enhance treatment accessibility through centralized procurement mechanisms that reduce high-cost medication expenditures, ultimately supporting progress toward the WHO's 2030 target for viral hepatitis elimination.

## Methods

### Type of economic evaluation

A cost-utility analysis was performed, considering direct medicine costs and health utility background. To evaluate the strategies, the analysis employed predicted numbers of liver-related diseases and deaths, costs (expressed in 2023 US dollars), quality-adjusted life-years (QALYs), and incremental cost-effectiveness ratios (ICERs). When the intervention demonstrates both cost-saving ($\Delta Cost < 0$) and superior effectiveness ($\Delta QALY > 0$), we will separately report incremental costs and QALYs to avoid misleading interpretation of negative ICER values

### Strategies

Three NAs antiviral therapies, endorsed by AASLD and CSH guidelines, were examined: ETV (0.5-mg tablet once daily), TDF (300-mg tablet once daily), and TAF (25-mg tablet once daily) [5,7]. Non-NAs best supportive care (BSC) constituted careful monitoring and essential therapy without antiviral treatment. Generic drugs were assumed to match brand-name drugs in quality and efficiency. All antiviral treatment strategies were compared to non-NAs BSC strategies.

Non-NAs BSC (Best Supportive Care without Nucleos(t)ide Analogues) includes:

Regular Monitoring: For patients in a stable phase, liver function tests (ALT, AST), HBV DNA quantification, and abdominal ultrasound are conducted every six months; for cirrhotic patients, monitoring is performed every three months.

Symptomatic Treatment: This involves the use of hepatoprotective medications (such as polyene phosphatidylcholine, glycyrrhizin preparations), diuretics for ascites management, and lifestyle interventions.

Guideline Adherence: Strict adherence to the follow-up standards outlined in the Chinese Guidelines for the Prevention and Treatment of Chronic Hepatitis B [7].

## Cohort and model structure

A Markov model simulated a hypothetical cohort of 100,000 Chinese CHB patients aged 14 years, meeting Chinese guideline criteria for hepatitis B treatment (Fig 1). The analysis employed TreeAge Pro Suite 2011 software (TreeAge Software, Inc., Williamstown, MA, USA). Patients transitioned across health states: CHB, compensated cirrhosis (CC), decompensated cirrhosis (DC), hepatocellular carcinoma (HCC), liver transplant (LT), post-liver transplant (PLT), and death. The model spanned a lifetime horizon of 64 years, aligning with the Chinese life expectancy of 78 years [35], The cycle length was 1 year. Refer to Table 1 for details on parameters used to derive transition probabilities between states, with related references. Principles for Determining Parameter Ranges:

For data supported by literature (the majority of data), ranges for parameters such as health state transition probabilities (e.g., CHB→cirrhosis), utility values, and complication costs are directly derived from the reported 95% confidence intervals (95% CI) or actual extremes from the literature; Handling Missing Data: Parameters that completely lack range data (such as liver transplant rates for liver cancer patients) or fixed parameters determined by policy (such as drug prices under NVBP policy) are included in the cost-utility analysis (CUA) as deterministic parameters. However, to validate the robustness of the CUA results, we set the parameter range from 0 to the fixed parameter value during sensitivity analysis to explore potential impacts on the model.

## Costs

Direct costs were calculated within a year from literature reports and a local hospital database, excluding indirect and intangible costs. Medical and non-medical expenses comprised the direct medical costs of disease states (outpatient and inpatient expenditures, self-purchased medicines), while direct non-medical costs encompassed travel expenses for treatment and additional health product expenses. Drug costs and monitoring expenses were evaluated based on local market

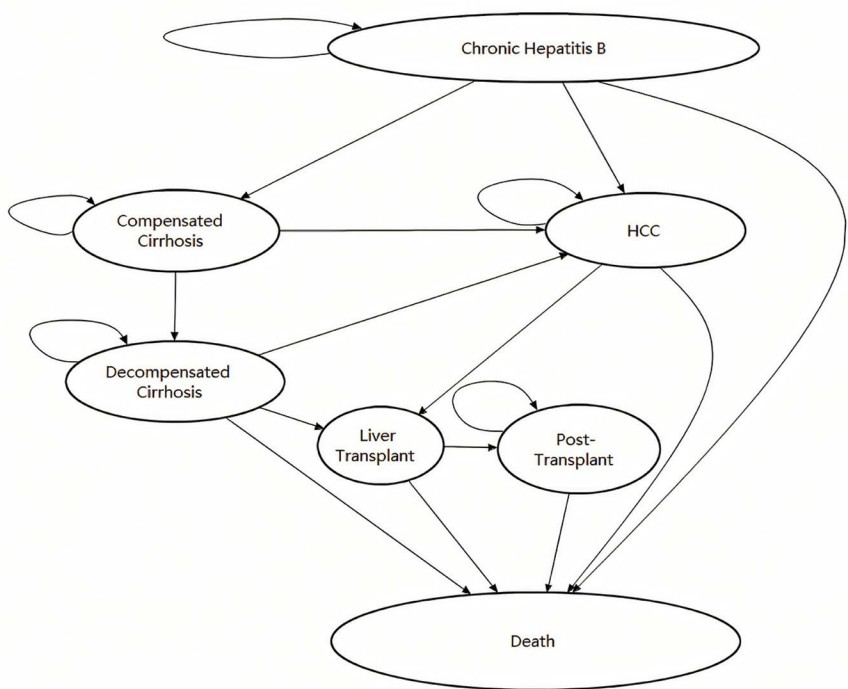

**Fig 1. State-transition model of chronic hepatitis B progression.**

**Table 1. Model inputs.**

| Parameter | Base-case | Range | Distribution | Source |
|---|---|---|---|---|
| **Annual transition probabilities** | | | | |
| CHB to CC | 1.90% | 1.0-2.8% | Beta | [37–41] |
| CHB to HCC | 0.59% | 0.37%−0.8% | Beta | [37,38,42,43] |
| CC to DC | 5.40% | 4.6-6.2% | Beta | [39–41,44] |
| CC to HCC | 4.27% | 1.94-6.6% | Beta | [39,43,45] |
| DC to HCC | 6.50% | 2.7-10.3% | Beta | [42,46–49] |
| Liver transplant of DC | 0.05%* | NA | NA | [2,4,23,50,51] |
| Death of DC | 9.55% | 9.0-10.1% | Beta | [48,49,52] |
| Liver Transplant of HCC | 0.20%* | NA | NA | [4,51] |
| Death of HCC | 29.03% | 27.45-30.60% | Beta | [53] |
| Death of Liver Transplant | 9.65% | 6.3%−13% | Beta | [54–56] |
| Death of Post-Liver Transplant | 6.12% | 3.98-8.25% | Beta | [57] |
| **HR of ETV vs. non-NA without Cirrhosis** | | | | |
| CHB to CC | 0.51 | 0.22-1.19 | lognormal | [12] |
| CHB to HCC | 0.40 | 0.28-0.57 | lognormal | [12] |
| **HR of ETV vs. non-NA with Cirrhosis** | | | | |
| CC to DC | 0.51 | 0.34-0.78 | lognormal | [12,58] |
| Cirrhosis to HCC | 0.55 | 0.31-0.99 | lognormal | [12,58] |
| Death or transplant of DC | 0.26 | 0.13-0.55 | lognormal | [12,58] |
| **HR of TDF vs. non-NA without Cirrhosis** | | | | |
| CHB to CC | 0.31 | 0.13-0.73 | lognormal | [12,59] |
| CHB to HCC | 0.27 | 0.07-0.98 | lognormal | [20,60,61] |
| **HR of TDF vs. non-NA with Cirrhosis** | | | | |
| CC to DC | 0.28 | 0.11-0.76 | lognormal | [59,62] |
| Cirrhosis to HCC | 0.46 | 0.29-0.75 | lognormal | [20,60,62] |
| Death or transplant of DC | 0.10 | 0.04-0.27 | lognormal | [62] |
| **HR of TAF vs. non-NA without Cirrhosis** | | | | |
| CHB to CC | 0.31** | 0.13-0.73 | lognormal | [12, 27, 63] |
| CHB to HCC | 0.24 | 0.06-0.87 | lognormal | [27, 60, 61] |
| **HR of TAF vs. non-NA with Cirrhosis** | | | | |
| CC to DC | 0.28** | 0.11-0.76 | lognormal | [27,28,64] |
| Cirrhosis to HCC | 0.41 | 0.26-0.67 | lognormal | [60,61] |
| Death or transplant of DC | 0.10** | 0.04-0.27 | lognormal | [27,28,64] |
| **Cost** | | | | |
| **Disease states(per year)** | | | | |
| CHB | 1739 | 870-2609 | Gamma | [26] |
| CC | 2955 | 1477-4432 | Gamma | [26] |
| DC | 5320 | 2661-7918 | Gamma | [26] |
| HCC | 18778 | 9389-28168 | Gamma | [26] |
| Liver transplant | 82001 | 62984-92467 | Gamma | [65] |
| Post-Liver Transplant | 9589 | 8219-10959 | Gamma | [66,67] |
| ETV | 1123 | NA | NA | [31] |
| ETV with NVBP | 9 | NA | NA | [32] |
| TDF | 1100 | NA | NA | [31] |
| TDF with NVBP | 15 | NA | NA | [32] |

*(Continued)*

**Table 1.** (Continued)

| Parameter | Base-case | Range | Distribution | Source |
|---|---|---|---|---|
| TAF | 1260 | NA | NA | [31] |
| TAF with NVBP | 47 | NA | NA | [32] |
| Surveillance of TDF adverse reactions | 72 | NA | NA | [9,32] |
| Surveillance of TAF adverse reactions | 39 | NA | NA | [9,32] |
| Discount Rate | 0.05 | NA | NA | [36] |
| **Utility scores** | | | | |
| CHB (EQ-5D) | 0.99 | 0.90-1.00 | Beta | [26] |
| CC (EQ-5D) | 0.80 | 0.70-0.90 | Beta | [26] |
| DC (EQ-5D) | 0.60 | 0.50-0.70 | Beta | [26] |
| HCC (EQ-5D) | 0.73 | 0.50-0.80 | Beta | [26] |
| Post-liver transplant (EQ-5D) | 0.84 | 0.77-0.91 | Beta | [33] |

CC compensated cirrhosis, CHB chronic hepatitis B, DC decompensated cirrhosis, HCC hepatocellular carcinoma, EQ-5D EuroQol five dimensions questionnaire, ETV entecavir, TDF tenofovir disoproxil fumarate, TAF tenofovir alafenamide, NVBP new volume-based purchasing

*Probability was calculated based on the liver-transplant cases and DC & HCC estimated numbers reported in a paper in 2011, China.

**Assume TAF equivalent to TDF

prices and data from Fujian Provincial Hospital [32]. All costs were converted from Chinese Yuan (CNY) to US dollars (USD) at an average exchange rate of 7.3:1 (CNY: USD) in 2023. Both costs and QALYs were discounted annually at a rate of 5%, following China's Guidelines for Pharmacoeconomic Evaluations [36].

## Willingness-to-pay (WTP) threshold

This study adopts the World Health Organization (WHO) recommendation of setting WTP thresholds at 1–3 times the per capita gross domestic product (GDP). Based on China's 2022 per capita GDP ($12,720.2), we established a WTP range of $12,500 to $37,500/QALY. The upper limit ($37,500/QALY) approximates 3 times the per capita GDP, aligning with WHO standards for middle-income countries and consistent with the Chinese Guidelines for Pharmacoeconomic Evaluations [36]. This threshold range spans decision-makers' acceptable cost-effectiveness ratios while ensuring analytical relevance to China's healthcare resource allocation realities.

## Utilities

Health state utilities were sourced from literature (Table 1), utilizing EuroQol Five Dimensions Questionnaire scores (EQ-5D). Scores ranged from 1 for perfect health to 0 for death.

## The cost-effectiveness frontier analysis

The cost-effectiveness frontier analysis is based on the absolute values of incremental costs and QALYs, generating an efficiency frontier by connecting non-inferior strategies (ICER≤WTP threshold). A strategy located below and to the right of the frontier line is considered dominant.

## Sensitivity Analyses

The model underwent sensitivity analyses encompassing parameters used within their specified ranges (Table 1). Monte Carlo simulation, comprising 100,000 iterations, assessed net monetary benefit (NMB) for multiple strategies and the Markov model, conducting probabilistic sensitivity analysis (PSA) to gauge the collective impact of parameter uncertainties on outcomes.

## Results

### Base-case results

A Monte Carlo microsimulation was executed using a 100,000-size trial cohort with parameters from Table 1. The outcomes, detailed in Table 2, encapsulate QALYs, life expectancy (LYs), incremental costs, incremental QALYs, and ICERs for the strategies assessed. Over a lifetime of therapy, compared to Non-NAs BSC, ETV, TDF, and TAF showcased significant reductions in deaths—25,345 (32.09%), 34,780 (44.04%), and 36,172 (45.80%), respectively (Table 2).

Without NVBP, Non-NAs BSC yielded the lowest life expectancy and QALYs (16.58 and 15.76 years) at a cost of 41,802.23 USD. Conversely, the TAF strategy achieved the highest life expectancy and QALYs (19.00 and 18.44 years) at a cost of 62,848.86 USD. ETV increased costs by 18,827.15 USD, offering 1.90 LYs and 2.01 QALYs incrementally, with an ICER of 9,366.74 USD/QALY. TDF increased costs by 18,623.38 USD, providing 2.34 LYs and 2.61 QALYs incrementally, with an ICER of 7,135.39 USD/QALY lower than ETV. Although TAF achieved the largest gains in LYs and QALYs (2.42 and 2.68 years), with an increased cost of 21,046.63 USD, its ICER stood at 7,853.22 USD/QALY, moderately higher than TDF.

With NVBP, costs for ETV, TDF, and TAF significantly decreased. Compared to non-NAs BSC, the lifetime costs for ETV(NVBP), TDF(NVBP), and TAF(NVBP) were even lower. Their ICERs—ETV(NVBP), TDF(NVBP), and TAF(NVBP)—showed values of −1,750.94 USD/QALY (cost-saving of 1,750.94 USD with 2.01 QALYs gained), −1,903.55 USD/QALY (cost-saving of 1,903.55 USD with 2.61 QALYs gained), and −1,998.25 USD/QALY (cost-saving of 1,998.25 USD with 2.68 QALYs gained), respectively. All three NVBP-negotiated regimens exhibited dominant cost-effectiveness profiles, achieving both cost savings and QALY gains compared to non-NAs BSC.

The World Health Organization recommends WTP/QALY ratios of 1–3 times the gross domestic product per capita [68]. With China's GDP per capita in 2022 at 12,720.2 USD [69], all strategies' ICERs were lower than one time the GDP per capita (Fig 2). The cost-effectiveness frontier analysis (Fig 2) shows that TAF(NVBP) is located at the far end of the efficiency frontier, indicating it is the only strategy achieving both cost savings (ΔCost=−1,998.25 USD/person) and the highest QALY gain (ΔQALY=2.68). Although TDF(NVBP) and ETV(NVBP) are also on the efficiency frontier, their QALY gains are lower than TAF(NVBP), and their costs are slightly higher (ΔCost of −1,903.55 USD and −1,750.94 USD, respectively).

### Fully incremental cost-effectiveness analysis

Moreover, we conducted a comprehensive fully incremental cost-effectiveness analysis separately under the conditions without NVBP and with NVBP (Table 3). Which systematically compares the incremental cost-effectiveness ratios (ICERs) for all strategy combinations. The results show that the ICER of TAF(NVBP) compared to TDF(NVBP) is −1,352.86 USD/

**Table 2. Base-case cost-effectiveness results.**

| Strategy | Cost (USD) | ΔCost (USD) | QALYs | ΔQALYs | ICER (USD/QALY) | LYs | ΔLYs | Death | ΔDeath | Dominance Status |
|---|---|---|---|---|---|---|---|---|---|---|
| Non-NAs BSC | 41802.23 | | 15.76 | | | 16.58 | | 78,972 | | Dominated |
| ETV | 60629.38 | 18827.15 | 17.77 | 2.01 | 9366.74 | 18.48 | 1.90 | 53,627 | −25,345 | Dominated |
| ETV(NVBP) | 40051.29 | −1750.94 | 17.77 | 2.01 | −871.11 | 18.48 | 1.90 | 53,627 | −25,345 | Dominated |
| TDF | 60425.61 | 18623.38 | 18.37 | 2.61 | 7135.39 | 18.92 | 2.34 | 44,192 | −34,780 | Dominated |
| TDF(NVBP) | 39898.68 | −1903.55 | 18.37 | 2.61 | −729.33 | 18.92 | 2.34 | 44,192 | −34,780 | Dominated |
| TAF | 62848.86 | 21046.63 | 18.44 | 2.68 | 7853.22 | 19.00 | 2.42 | 42,800 | −36,172 | Dominated |
| TAF(NVBP) | 39803.98 | −1998.25 | 18.44 | 2.68 | −745.62 | 19.00 | 2.42 | 42,800 | −36,172 | Dominant |

Non-NAs BSC best supportive care without nucleos(t)ide analogues, ETV entecavir, TDF tenofovir disoproxil fumarate, TAF tenofovir alafenamide, NVBP new volume-based purchasing, ICER incremental cost-effectiveness ratio, QALY quality-adjusted life-year, LY life year expectancy

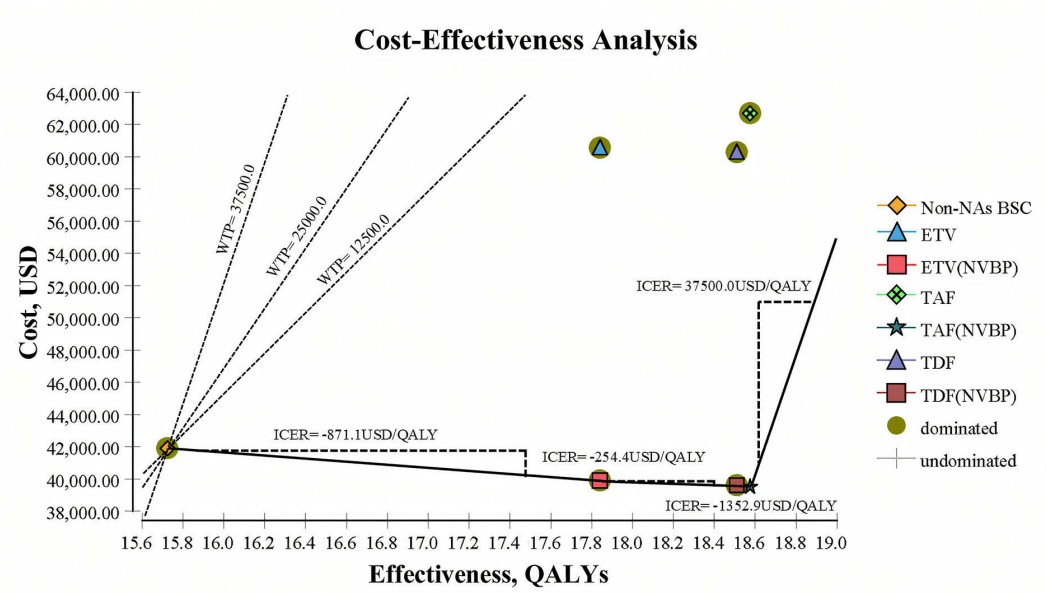

**Fig 2. Cost-effectiveness Frontier plot of various treatments for chronic hepatitis B patients.** The x-axis represents the lifetime quality-adjusted life years gained for each therapy, and the y-axis indicates the lifetime costs (US dollar). Non-NAs BSC served as a comparator. Three dashed lines through the BSC strategy point represent the range of Chinese willingness-to-pay (WTP) thresholds (12,500–37,500 USD/QALY). The solid line connecting dominant strategies forms the efficiency frontier, with the theoretical boundary at 37,500 USD/QALY shown by the solid line extending to the right from TAF(NVBP). Non-NAs BSC best supportive care without nucleos(t)ide analogues, ETV entecavir, TDF tenofovir disoproxil fumarate, TAF Tenofovir alafenamide, NVBP New Volume-based purchasing, WTP willingness-to-pay (USD/QALY).

QALY, indicating that TAF(NVBP) is a dominant strategy with lower costs and higher health outcomes. The ICER of TAF(NVBP) compared to ETV(NVBP) is −369.12 USD/QALY, also showing dominance. The ICER of TDF(NVBP) compared to ETV(NVBP) is −254.35 USD/QALY, further confirming the ranking of strategies. In probabilistic sensitivity analysis (PSA), TAF(NVBP) was superior to TDF(NVBP) in 54.23% of simulations (Fig 5C) and superior to ETV(NVBP) in 86.24% of simulations (Fig 5B), establishing a clear cost-effectiveness hierarchy: TAF(NVBP)> TDF(NVBP)> ETV(NVBP).

Comparison with Baseline Strategy, at a willingness-to-pay (WTP) threshold of 37,500 USD/QALY, the ICER of TAF(N-VBP) compared to non-NAs BSC is −745.62 USD/QALY, indicating both cost savings and health gains (Table 2). The fully incremental analysis further confirms the optimal position of TAF(NVBP) among all strategies (Table 3); even when the WTP threshold is reduced to 12,500 USD/QALY, the probability of it being the optimal strategy remains above 51% (Fig 6).

### Tornado diagram and Univariate Sensitivity analysis

The tornado diagram analyzed sensitivity across transition probabilities, costs, and utilities. In the model, the age of therapy initiation had the most substantial impact on NMB over a lifetime, followed by the discount rate. Initiating therapy at 14 years led to a gain of 654,941.93 USD in NMB, contrasting with 289,268.18 USD at 77 years indicating that earlier initiation results in greater NMB. Additionally, with a discount rate increase from 3% to 5%, NMB decreased from 923,183.53 USD to 654,941.93 USD. These two parameters contributed 96.7% cumulative risk together.

Parameters related to treatment strategies, such as hazard ratios (HR) or treatment costs, The HR of CHB to CC and CHB to HCC with TAF were relatively crucial. However, variations in these parameters only caused a marginal NMB change (−0.88% to 2.16%) around the expected value of 656,839.29 USD. Consequently, apart from therapy initiation age and the discount rate, other parameters marginally affected model robustness (Fig 3).

**Table 3. Results of the fully incremental cost-effectiveness analysis.**

| Strategy | Cost (USD) | Effectiveness(QALY) | Incremental costs(USD) | Incremental effectiveness (QALY) | ICER (USD/QALY) |
|---|---|---|---|---|---|
| TAF(NVBP) | 39803.98 | 18.44 | | | |
| TDF(NVBP) | 39898.68 | 18.37 | | | |
| ETV(NVBP) | 40051.29 | 17.77 | | | |
| TAF | 62848.86 | 18.44 | | | |
| TDF | 60425.61 | 18.37 | | | |
| ETV | 60629.38 | 17.77 | | | |
| Non-NAs BSC | 41802.23 | 15.76 | | | |
| With NVBP | | | | | |
| TAF vs. TDF | | | −94.7 | 0.07 | −1352.857143 |
| TAF vs. ETV | | | −247.31 | 0.67 | −369.119403 |
| TDF vs. ETV | | | −152.61 | 0.6 | −254.35 |
| TAF vs. Non-NAs BSC | | | −1998.25 | 2.68 | −745.6156716 |
| TDF vs. Non-NAs BSC | | | −1903.55 | 2.61 | −729.3295019 |
| ETV vs. Non-NAs BSC | | | −1750.94 | 2.01 | −871.1144279 |
| Without NVBP | | | | | |
| TAF vs. TDF | | | 2423.25 | 0.07 | 34617.86 |
| TAF vs. ETV | | | 2219.48 | 0.67 | 3312.657 |
| TDF vs. ETV | | | −203.77 | 0.6 | −339.617 |
| TAF vs. Non-NAs BSC | | | 21046.63 | 2.68 | 7853.22 |
| TDF vs. Non-NAs BSC | | | 18623.38 | 2.61 | 7135.395 |
| ETV vs. Non-NAs BSC | | | 18827.15 | 2.01 | 9366.741 |

Non-NAs BSC best supportive care without nucleos(t)ide analogues, ETV entecavir, TDF tenofovir disoproxil fumarate, TAF tenofovir alafenamide, NVBP new volume-based purchasing, ICER incremental cost-effectiveness ratio, QALY quality-adjusted life-year

## Probabilistic sensitivity analysis

The PSA, employing 100,000 Monte Carlo simulations, assessed the uncertainties' impact. In comparisons without NVBP, ETV, TDF, and TAF demonstrated superiority or acceptable ICERs in 99.12% (Fig 4A), 99.77% (Fig 4B), and 99.85% (Fig 4C) of simulations compared to non-NAs BSC, respectively, at a WTP threshold of 37,500 USD/QALY. In NAs-related comparisons, TDF and TAF showed superiority or acceptable ICERs in 83.44% (Fig 4D) and 84.34% (Fig 4E) of simulations, respectively, compared to ETV at the same WTP threshold. At the same time, the TAF only shows superiority or acceptable ICER in 49.88% (Fig 4F) simulation compared to TDF at the WTP threshold of 37,500 USD.

With NVBP, TDF(NVBP) and TAF(NVBP) maintained superiority or acceptable ICERs in most simulations against ETV(NVBP) (82.96% and 86.24% at 37,500 USD WTP, Fig 5A & 5B). Remarkably, TAF(NVBP) showcased superiority over TDF(NVBP) in over half of simulations (54.23% at 37,500 USD WTP, Fig 5C), a scenario not observed without NVBP. Notably, TDF without NVBP failed to show superiority in most simulations compared to TAF(NVBP) (14.58%, Fig 5D), and TAF without NVBP failed to show superiority in most simulations compared to TDF(NVBP) (17.63%, Fig 5E) either at 37,500 USD WTP. The TAF(NVBP) consistently exhibited superiority over non-NAs BSC in nearly all simulations (99.96% at 37,500 USD, Fig 5F).

Although probabilistic sensitivity analysis (PSA) demonstrated the model's robustness to most parameters, we conducted scenario analyses to further validate model stability. These included scenarios such as shortening the antiviral treatment duration to 40 or 20 years, adjusting discount rates to 3% or 7%, and varying monitoring costs by ±50%. The results indicated that TAF (NVBP) remained the optimal strategy across all examined scenarios (S1 Fig).

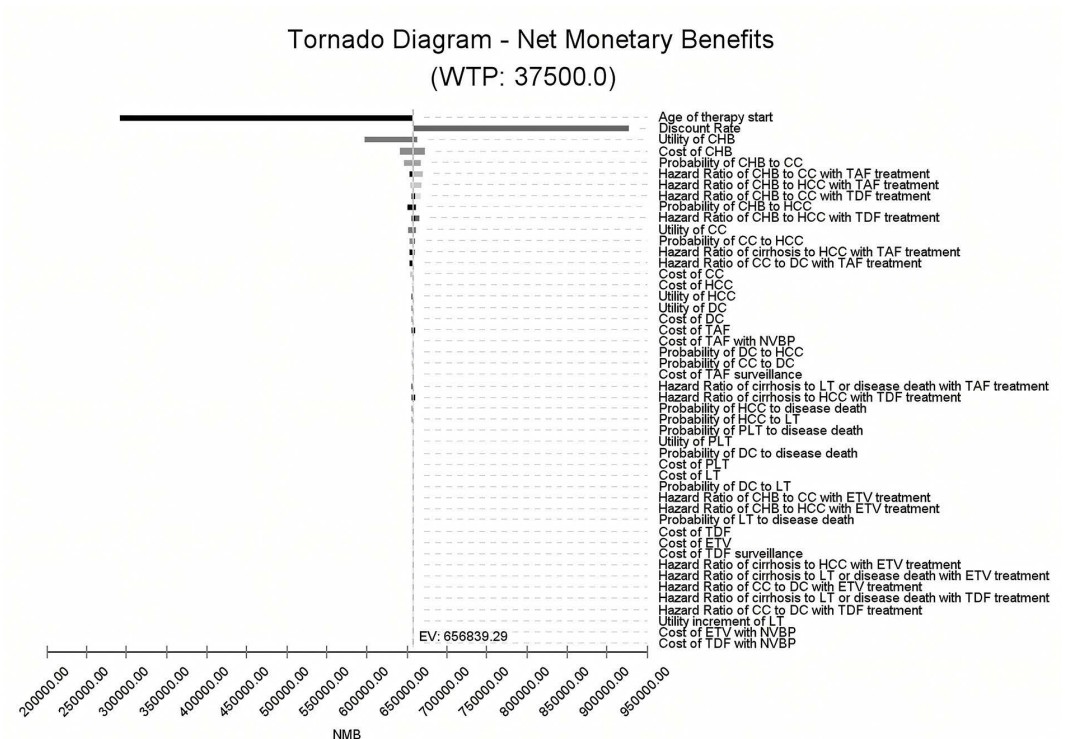

**Fig 3. Univariate sensitivity analyses (net monetary benefit).** Tornado diagram comparing univariate sensitivity analyses of all parameters on the cost-effectiveness of strategies in the chronic hepatitis B model at a threshold of 37,500 USD/quality-adjusted lifer-year gained. CC compensated cirrhosis, CHB chronic hepatitis B, DC decompensated cirrhosis, HCC hepatocellular carcinoma, LT liver transplant, HR hazard ratio, EV expected value, NMB net monetary benefit, ETV entecavir, TDF tenofovir disoproxil fumarate, TAF Tenofovir alafenamide, NVBP New Volume-based purchasing.

The cost-effectiveness acceptability curve revealed that TAF(NVBP) had a 52.47% chance of being the optimal treatment at a WTP threshold of 37,500 USD/QALY, followed by TDF(NVBP) and ETV(NVBP) at 43.09% and 4.48%, respectively (Fig 6). Even at lower WTP thresholds, TAF(NVBP) retained over half the probability of being the most cost-effective treatment for CHB infections (52.11% at 25,000 USD WTP and 51.15% at 12,500 USD WTP, Fig 6).

## Discussion

In this study, we conducted a comprehensive evaluation of the cost-effectiveness of first-line NAs for lifelong CHB therapy, specifically focusing on ETV, TDF, and TAF within the NVBP framework. Our analyses consistently demonstrated that, in comparison to the non-NAs BSC strategy, all NAs, irrespective of NVBP, exhibited cost-effectiveness across a range of WTP thresholds from 12,500 USD/QALY to 37,500 USD/QALY.

Without NVBP, TAF demonstrated the highest QALYs gain despite incurring the highest drug cost. TAF consistently exhibited superiority or acceptable ICERs compared to ETV in the majority of simulations. However, even considering the additional surveillance required for TDF treatment, TAF only demonstrated superiority or acceptable ICER in less than half of the simulations. In contrast, under NVBP, despite TAF(NVBP) maintaining the highest QALYs gain and the highest drug cost, it consistently exhibited superiority or acceptable ICER in most simulations compared to TDF(NVBP) and ETV(N-VBP). TAF(NVBP) emerges as the dominant strategy by achieving maximum QALY gains while reducing costs. Given the acknowledged improvements in renal and bone safety associated with TAF [27], our findings suggest that TAF(NVBP)

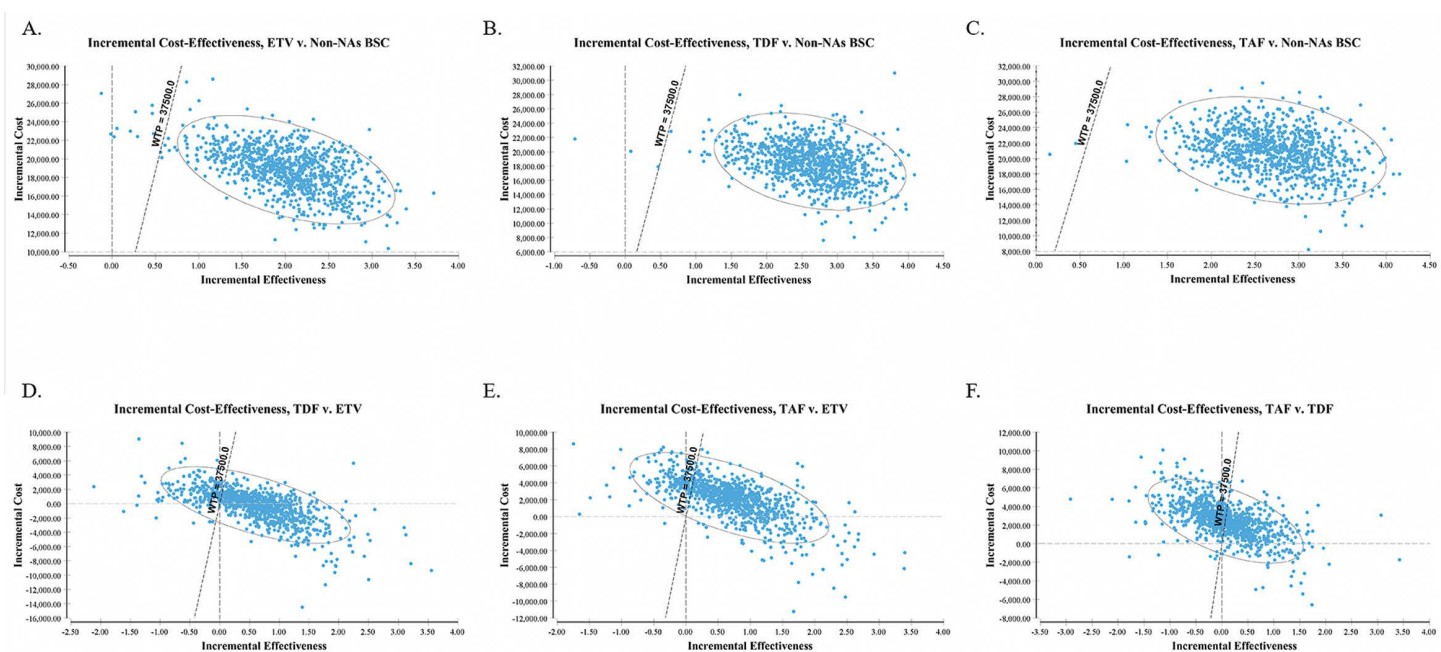

**Fig 4. Probabilistic results of the incremental cost-effectiveness pairwise between various treatments without NVBP for a simulation of 100,000 iterations.** The y-axis represents the incremental costs. The x-axis represents the incremental quality-adjusted life years (QALYs) gained. BSC best supportive care, ETV entecavir, TDF tenofovir disoproxil fumarate, TAF Tenofovir alafenamide.

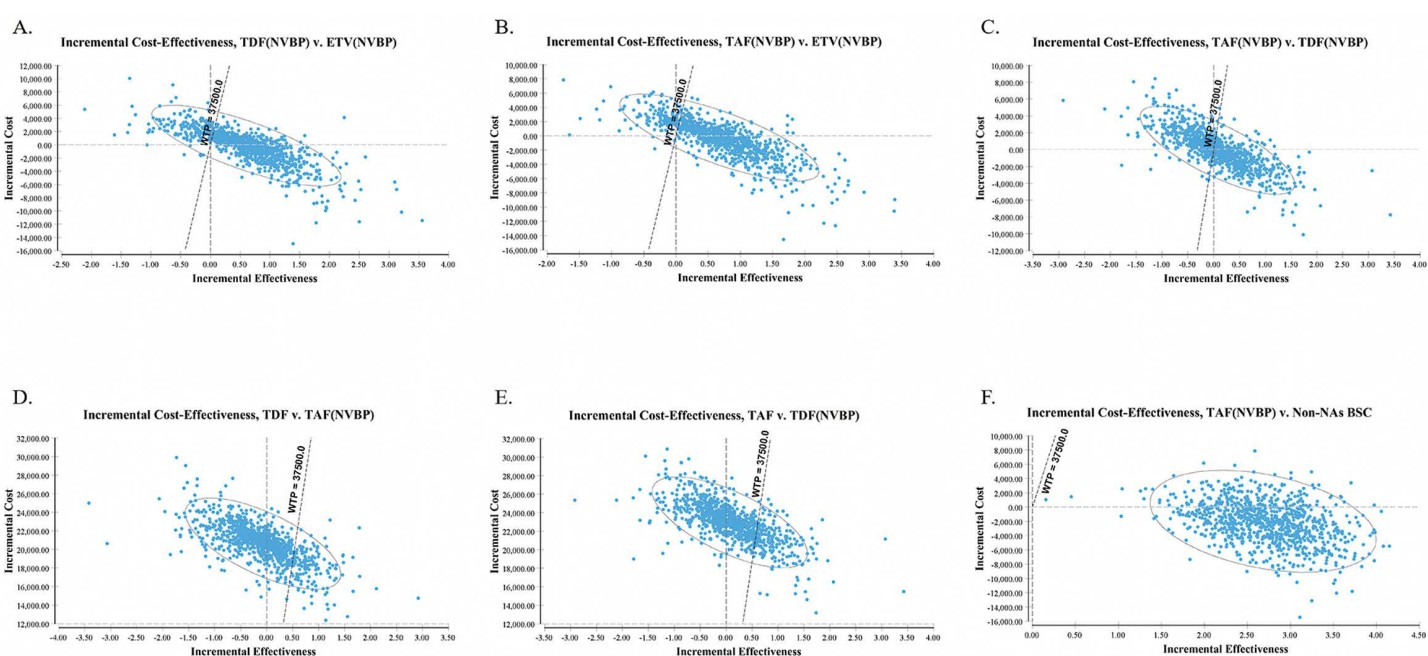

**Fig 5. Probabilistic results of the incremental cost-effectiveness pairwise between various treatments with NVBP for a simulation of 100,000 iterations.** The y-axis represents the incremental costs. The x-axis represents the incremental quality-adjusted life years (QALYs) gained. Non-NAs BSC best supportive care without nucleos(t)ide analogues, ETV entecavir, TDF tenofovir disoproxil fumarate, TAF Tenofovir alafenamide, NVBP New Volume-based purchasing.

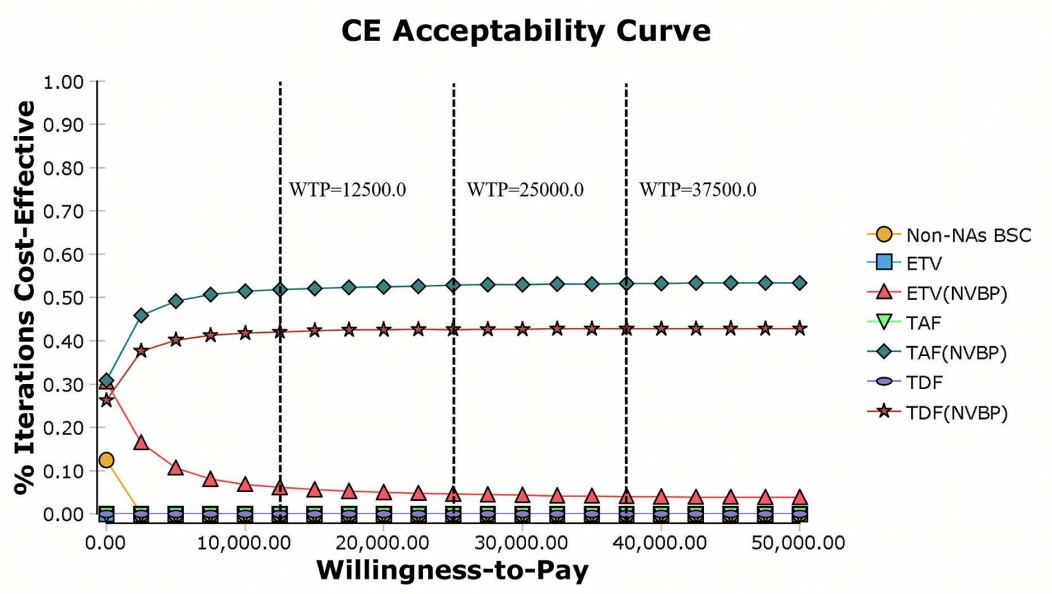

**Fig 6. Probabilistic sensitivity analysis: cost-effectiveness acceptability curve.** Non-NAs BSC best supportive care without nucleos(t)ide analogues, ETV entecavir, TDF tenofovir disoproxil fumarate, TAF Tenofovir alafenamide, NVBP New Volume-based purchasing, WTP willingness-to-pay (USD/QALY).

stands out as the optimal therapy across all strategies, both with and without NVBP, within a WTP threshold ranging from 12,500–37,500 USD/QALY.

A study by Tian et al. indicated that TAF is not cost-effective compared to ETV and TDF in Canada, suggesting a 33.4% reduction in price for TAF to achieve cost-effectiveness [33]. In our Chinese context, where brand prices for ETV, TDF, and TAF without NVBP were notably lower (28.7%, 21.6%, and 77.5%, respectively) than the prices reported by Tian, the increased cost-effectiveness of TAF in China is plausible. With NVBP and annual price refreshing, the costs of ETV(NVBP), TDF(NVBP), and TAF(NVBP) in our hospital substantially decreased (the price of units of ETV, TDF, and TAF in our hospital were 0.03 USD, 0.05 USD, and 0.13 USD in 2023 [32]) leading to their dominance over ETV, TDF, and TAF in our analyses. Even considering TAF's highest QALYs gain, the cost-effectiveness analysis demonstrated that TAF(NVBP) would be the optimal strategy for most individuals, followed by TDF(NVBP) and ETV(NVBP). TAF, without NVBP, did not exhibit clear superiority to TDF.

Our analysis assumes generic drugs are equivalent in potency and quality to brand-name drugs [70], as NVBP is open to both generic and brand-name drug manufacturers. China's National Medical Products Administration (NMPA) mandates strict bioequivalence (BE) standards through its Generic Consistency Evaluation (GCE) framework. All generic drugs procured via the National Volume-Based Procurement Policy (NVBP) must demonstrate BE, requiring geometric mean ratios (GMRs) of pharmacokinetic parameters (AUC and Cmax) to fall within 80%−125% compared to reference products. For instance, TAF generics have shown GMRs of 98.7%−103.2% (Center for Drug Evaluation Annual Report, 2022) [71], confirming short-term bioequivalence. Real-world evidence (e.g., Hou et al., 2021) further supports this equivalence, showing comparable virological response rates (HBV DNA<20 IU/mL) between generic and originator TAF (72.3% vs. 71.8%, p=0.62) [27], validating our assumption of therapeutic equivalence. However, long-term clinical outcomes (e.g., cirrhosis or hepatocellular carcinoma incidence) under the GCE framework remain understudied. While NMPA requires post-marketing surveillance by generic manufacturers, direct comparisons of long-term efficacy and safety between generics and originators are currently unavailable. Therefore, future research should leverage NVBP's real-world data platforms to

continuously validate generics' long-term performance, particularly their preventive effects against end-stage liver disease events.

ETV, TDF, and TAF effectively suppress HBV DNA replication, notably benefiting CHB outcomes and preventing HCC occurrence [12–15]. However, none eliminate cccDNA, and HBV viremia typically recurs post-therapy cessation despite successful virus suppression [10]. Long-term NAs therapy has shown promise in reducing mortality and HCC incidence in CHB patients justifying our modeling of a lifetime therapy scale [12,15,16,72–74]. As expected, NAs therapy's costs exceeded non-NAs BSC but were cost-effective based on ICERs. Interestingly, under NVBP, NAs' whole life costs were lower than non-NAs BSC, possibly due to prognosis improvements.

Recent clinical studies and real-world data further substantiate the cost-effectiveness of TAF under NVBP. A multicenter cohort study by Hou et al. (2021) [75] demonstrated that TAF achieved comparable virological suppression to TDF while significantly reducing renal dysfunction (3.2% vs. 8.7%, p < 0.01) and bone density loss (1.5% vs. 6.4%, p < 0.01) over a 3-year follow-up in Chinese CHB patients. These safety advantages align with our model's assumption of reduced long-term monitoring costs for TAF.

Additionally, Li et al. (2020) [19] conducted a meta-analysis showing that TDF was associated with a 22% lower risk of HCC compared to ETV (adjusted HR = 0.78, 95% CI:0.64–0.95) in cirrhotic patients, supporting our finding that TAF (as a safer alternative to TDF) may further enhance QALY gains. Notably, the implementation of NVBP has been shown to improve treatment adherence. For instance, Yuan et al. (2021) [70] reported that drug price reductions under NVBP increased adherence by 34% in chronic disease populations, which likely enhances viral suppression and reduces decompensation events, as reflected in our probabilistic sensitivity analysis (PSA) results.

Further evidence from Jeong et al. (2022) [28] in a real-world Korean cohort confirmed that TAF maintains similar efficacy to TDF while mitigating renal and bone risks, thereby reducing hospitalization costs by 18%—a key factor in our cost-saving calculations. These clinical and economic synergies reinforce our conclusion that TAF(NVBP) represents the optimal strategy for CHB management in resource-limited settings.

Several limitations identified in our study call for further refinement in future research. Our consideration of surveillance costs for TDF was limited to conforming with the Asian-Pacific clinical practice guidelines. Studies, such as Kosh et al.'s research, revealed a decrease in bone mineral density over a 96-week TDF treatment period, yet without any occurrence of osteoporosis among patients [76]. Similarly, Shun et al.'s findings indicated that renal dysfunction attributed to TDF could be rectified by switching to TAF without additional treatment [77]. These studies suggest that the costs associated with TDF mainly stem from surveillance rather than direct medical intervention. Additionally, the assumption of equivalence in efficacy between TDF and TAF, albeit supported by various studies and guidelines, requires further validation. Studies like Base Lee's research, which found no significant difference in the risk of HCC development between TDF and TAF groups of CHB patients [61], and Jeoong's investigation demonstrating similar antiviral effects of both drugs over 48 weeks [28], contribute to this assumption. Furthermore, Hou's study conducted among chronic hepatitis B patients from China revealed that TAF treatment offered efficacy similar to TDF but with improved renal and bone safety over three years [27]. The AASLD 2018 guidelines also suggest that TAF presents a superior safety profile compared to TDF, with similar antiviral efficacy observed in studies spanning up to two years [21]. Nevertheless, to solidify these assertions, future long-term comparative studies directly comparing the efficacy of TDF and TAF are warranted.

China's per capita healthcare expenditure stood at 4,702.8 CNY (737.9 USD) in 2019 [78], notably lower than the U.S. figure of 11,702.41 USD in 2020 [79]. Additionally, the prevalence of HBsAg is significantly higher in China with 84 million affected individuals compared to 1.59 million in the United States [80]. This stark contrast in healthcare spending and HBsAg prevalence underscores the critical need for cost-effective, long-term Nucleos(t)ide Analogs (NAs) for Chronic Hepatitis B (CHB) patients in China. Our analysis revealed that ETV(NVBP), TDF(NVBP), and TAF(NVBP) strategies exhibited lower costs compared to Non-NAs BSC, with TAF(NVBP) showing the highest acceptability probability. TAF with

NVBP emerges as a pivotal option in alleviating the burden of CHB, presenting as an optimal strategy without the concomitant escalation of healthcare expenditure.

## Conclusion

Our analysis suggests that TAF with NVBP represents the most cost-effective long-term therapy for CHB. Both TDF and ETV, with or without NVBP, and TAF without NVBP were considered cost-ineffective.

## Supporting information

**S1 Fig. Cost-effectiveness scenario analysis of various therapeutic regimens for chronic hepatitis B.** (PDF)

## Acknowledgments

None.

## Author contributions

**Conceptualization:** Yi Lin, Xueyan Lin, Zhihui Lin, Shiyun Lu.

**Data curation:** Yi Lin, Xueyan Lin, Juan Chen, Ruiqi Xia.

**Investigation:** Juan Chen.

**Methodology:** Yi Lin.

**Project administration:** Yi Lin, Shiyun Lu.

**Supervision:** Ruiqi Xia, Zhihui Lin, Shiyun Lu.

**Writing – original draft:** Yi Lin, Xueyan Lin.

**Writing – review & editing:** Zhihui Lin, Shiyun Lu.

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
