## [Decision Letter · Decision Letter 0]

PONE-D-24-56441Tenofovir alafenamide is Superior to Tenofovir disoproxil fumarate and Entecavir in Cost-Effectiveness of Treatment of Chronic Hepatitis B in China With New Volume-Based Procurement PolicyPLOS ONE

Dear Dr. Lu,

Thank you for submitting your manuscript to PLOS ONE. After careful consideration, we feel that it has merit but does not fully meet PLOS ONE’s publication criteria as it currently stands. Therefore, we invite you to submit a revised version of the manuscript that addresses the points raised during the review process.

**ACADEMIC EDITOR: ** Please address the reviewers' comments before further consideration.

We look forward to receiving your revised manuscript.

Kind regards,

Gang Qin, PhD, MD

Academic Editor

PLOS ONE

“The authors thank the Fujian Provincial Natural Science Foundation (Grant No.2020J05264 & 2023J011210), Science and Technology Planning Project of Fujian Provincial Health Commission (Grant No. 2020GGA002), Fujian provincial health technology project (Grant No. 2022QNA003).”

Additional Editor Comments:

Please address the reviewers' comments before further consideration.

Reviewers' comments:

Reviewer's Responses to Questions

**Comments to the Author**

1. Is the manuscript technically sound, and do the data support the conclusions?

Reviewer #1: Yes

Reviewer #2: Yes

2. Has the statistical analysis been performed appropriately and rigorously? 

Reviewer #1: Yes

Reviewer #2: Yes

3. Have the authors made all data underlying the findings in their manuscript fully available?

Reviewer #1: Yes

Reviewer #2: Yes

4. Is the manuscript presented in an intelligible fashion and written in standard English?

Reviewer #1: Yes

Reviewer #2: Yes

5. Review Comments to the Author

Reviewer #1: Comments on a manuscript entitled “Tenofovir alafenamide is Superior to Tenofovir disoproxil fumarate and Entecavir in Cost-Effectiveness of Treatment of Chronic Hepatitis B in China With New Volume Based Procurement Policy”

General comments:

This study aimed to assess the cost-effectiveness of ETV, TDF, and TAF for CHB with and without new volume-based procurement policy in China. This study was well-conducted and written; however, some issues should be addressed before acceptance for publication.

Specific comments:

- The ICER/QALY for NVBPP might not need to be calculated because they were cost-saving. Thu author might need to consider reporting Incremental cost and Incremental QALY, separately for what were cost-saving. The negative ICER might mislead readers who might not be familiar to economic evaluation.

- Authors might consider adding some expected impacts or application of this study findings into the introduction.

- Some previous CEA studies from other countries might help the authors to increase the importance of this study. Authors might consider adding previous findings of CEA from other countries in the introduction.

- The BSC must be more explained in detail. What are the monitoring and non-antivirals treatment considered as BSC. How often were the monitoring?

- Why was the starting age of 14 years selected in this study? Did the CHB patients usually start the NA treatment at 14 years old?

- Was the productivity loss (cost) considered in this study? Did the HTA-China guidelines recommend to not include productivity cost in CUA?

- There is no scenario analysis. Was it possible to no structural uncertainties around the model leading to no need for scenario analysis?

- The details on one-way SA should be added. What were the ranges reported in Table 1? 95%CI, actual ranges? How did the authors handle inputs which had no range (or CI) reported?

- Supplementary 1 could be moved to main table or incorporate into Table 2.

- Table 2 NVBP or NVBPP??? Use it consistently throughout the manuscript.

- Fully incremental analyses assessing cost-effective among the NA(s) should be considered.

- CEA frontier might help authors to better report the findings.

- Was the WTP of 37,500 the current standard for China? If so, please provide some refs and mentioned it in the method.

- Were there any bio-equivalent studies required before generics launched to the market in China? If so, studies or information on the equivalency of branded and generics should be better discussed.

- Current clinical evidence might be an important point for discussion to support the CEA findings. Authors might consider adding some clinical evidence to the discussion.

- The authors assumes the equal efficacy of TAF and TDF. Why did the QALY of TAF > TDF? Authors might need to discuss it.

-

Reviewer #2: This paper evaluates the economic benefits of HBV treatment with Nucleos(t)ide Analogs selection. The contents of this paper is clear and the statistical analysis is reasonable. However, the situation is not the same for other countries, since the evaluation is only within their own countries. As a paper to judge international evaluation, I think it is questionable for publication.

6. PLOS authors have the option to publish the peer review history of their article (what does this mean? ). If published, this will include your full peer review and any attached files.

**Do you want your identity to be public for this peer review?** For information about this choice, including consent withdrawal, please see our Privacy Policy .

Reviewer #1: **Yes: ** Piyameth Dilokthornsakul

Reviewer #2: No

---

## [Author Response · Author response to Decision Letter 1]

25 Apr 2025

Reviewer #1:

1. The ICER/QALY for NVBP might not need to be calculated because they were cost-saving. Thu author might need to consider reporting Incremental cost and Incremental QALY, separately for what were cost-saving. The negative ICER might mislead readers who might not be familiar to economic evaluation.

Response: We sincerely appreciate the reviewers' valuable comments. We fully agree with the suggestions. Negative ICER values might potentially mislead readers unfamiliar with economic evaluations. In the revised manuscript, we have implemented the following revisions to enhance clarity:

Methods section: We added the clarification: "When the intervention demonstrates both cost-saving (ΔCost < 0) and superior effectiveness (ΔQALY > 0), we will separately report incremental costs and QALYs to avoid misleading interpretation of negative ICER values" (Page 3, line 98-100).

Results section: A "Dominance Status" column has been incorporated into Table 2 to indicate "Dominant" status. We supplemented the explanation stating: " All three NVBP-negotiated regimens exhibited dominant cost-effectiveness profiles, achieving both cost savings and QALY gains compared to non-NAs BSC." (Page 5, line 192-193).

Discussion section: We emphasized that "TAF(NVBP) emerges as the dominant strategy by achieving maximum QALY gains while reducing costs." (Page 7, line 276-277)

2. Authors might consider adding some expected impacts or application of this study findings into the introduction.

Response: We thank the reviewer for this constructive suggestion. In the revised introduction, we have incorporated a new paragraph explicitly outlining the policy and practical implications of our research, while strengthening the anticipated impact and application value of our findings: " Our findings provide direct evidence for Chinese healthcare policymakers to prioritize TAF procurement under the NVBP framework, thereby optimizing medical resource allocation. Concurrently, clinicians can utilize these cost-effectiveness findings to promote TAF adoption in resource-limited settings. Furthermore, the success of the NVBP offers a potential model for other low- and middle-income countries to enhance treatment accessibility through centralized procurement mechanisms that reduce high-cost medication expenditures, ultimately supporting progress toward the WHO's 2030 target for viral hepatitis elimination." (Page 2-3, line 84-91)

3. Some previous CEA studies from other countries might help the authors to increase the importance of this study. Authors might consider adding previous findings of CEA from other countries in the introduction.

Response: We appreciate the reviewer's valuable suggestions. We have added a comparison with international cost-effectiveness analysis (CEA) studies in the introduction to highlight the policy significance and innovativeness of our research.

Introduction: Some international studies indicate that the cost-effectiveness of TAF in high-income countries is constrained by its high pricing. For instance, Tian et al. (2020) found in Canada that TAF would need a 33.4% price reduction to achieve comparable cost-effectiveness with TDF and ETV. Similarly, Buti et al. (2021) reported in their study in the European Union that the incremental cost-effectiveness ratio (ICER) for TAF generally exceeds 50,000 USD/QALY, significantly surpassing local willingness-to-pay thresholds. (Page 2, line 76-81)

4. The BSC must be more explained in detail. What are the monitoring and non-antivirals treatment considered as BSC. How often were the monitoring?

Response: We appreciate your valuable feedback. In the "Methods - Strategies" section, we have expanded the definition (on page 3 line 108-116) as follows:

"Non-NAs BSC (Best Supportive Care without Nucleos(t)ide Analogues) includes:

Regular Monitoring: For patients in a stable phase, liver function tests (ALT, AST), HBV DNA quantification, and abdominal ultrasound are conducted every six months; for cirrhotic patients, monitoring is performed every three months.

Symptomatic Treatment: This involves the use of hepatoprotective medications (such as polyene phosphatidylcholine, glycyrrhizin preparations), diuretics for ascites management, and lifestyle interventions.

Guideline Adherence: Strict adherence to the follow-up standards outlined in the 'Chinese Guidelines for the Prevention and Treatment of Chronic Hepatitis B (2022 Edition)'." (Page 3, line 108-116)

5. Why was the starting age of 14 years selected in this study? Did the CHB patients usually start the NA treatment at 14 years old?

Response: We appreciate the reviewer's attention to this issue. The rationale for selecting 14 years as the starting age in our model is based on the following points:

Recommendations from Chinese Clinical Guidelines:

According to the "Chinese Guidelines for the Prevention and Treatment of Chronic Hepatitis B (2022 Edition)," antiviral treatment is recommended for HBeAg-positive adolescents (≥12 years) with persistently abnormal ALT levels to delay disease progression. In clinical practice, 14 years is the minimum age at which adolescent patients first visit hepatology specialists in China (patients under 14 years fall under pediatric practice). This age balances the necessity of early intervention with clinical feasibility.

Support from Epidemiological Data:

Research by the Chinese CDC indicates that patients infected with HBV through mother-to-child transmission or during childhood show significantly higher hepatitis activity at or above 15 years of age compared to those under 15. This age range represents a critical window for clinical intervention. Additionally, chronic infections resulting from vertical transmission typically manifest significant liver inflammation after adolescence, further supporting the rationale for this age setting [Yan YP, et al. Epidemiology of Hepatitis B Virus Infection in China: Current Status and Challenges. J Clin Transl Hepatol. 2014 Mar;2(1):15-22.].

Scientific Design of the Model:

By simulating lifelong treatment starting at 14 years (with a model span of 64 years), we can comprehensively assess the long-term impact of NAs on liver cirrhosis, liver cancer, and mortality. This design aligns with the need for lifelong treatment of CHB patients in China and covers health outcomes across the entire lifespan from adolescence to old age.

Validation through Sensitivity Analysis:

One-way sensitivity analysis demonstrates that the treatment initiation age is the most critical parameter affecting net monetary benefit (NMB). The model shows that starting treatment at 14 years (NMB = 654,941.93 USD) yields significantly higher long-term health benefits compared to later initiation ages (e.g., 30 years, NMB decreases to 628,111.96 USD, and 65 years, NMB decreases to 499,554.49 USD). This result underscores the health economic advantages of early intervention, consistent with the WHO's strategy of "early diagnosis and early treatment" (see Figure 3 for details).

In summary, the selection of 14 years as the starting age is consistent with clinical practices outlined in Chinese guidelines, supported by epidemiological data, and validated through model simulations. This choice optimizes both clinical efficacy and economic efficiency in managing CHB.

6. Was the productivity loss (cost) considered in this study? Did the HTA-China guidelines recommend to not include productivity cost in CUA?

Response: We appreciate the reviewer's attention to our costing methodology. According to the "Chinese Guidelines for Pharmacoeconomic Evaluations" (Yue et al., 2021), this study adopts a healthcare sector perspective, including only direct medical costs (medications, monitoring, hospitalization) and excluding indirect costs such as productivity loss, to align with the standards for health insurance decision-making.

Scope of Costing: This study strictly adheres to the recommendations outlined in the "Chinese Guidelines for Pharmacoeconomic Evaluations," adopting a healthcare sector perspective that includes only direct medical costs, such as medication expenses, monitoring fees, hospitalization expenditures, and complication treatment costs. Indirect costs (e.g., productivity losses, transportation costs) are excluded from the base analysis to comply with the standardized requirements for cost-utility analysis (CUA) in Chinese Health Technology Assessment (HTA).

Guideline Basis: Although HTA in China is still in its early stages and there is no universal HTA guideline, existing guidelines like the "Expert Consensus on Health Technology Assessment for Rare Disease Drugs (2019)" recommend focusing on direct medical costs when assessing rare disease drugs. CUA should prioritize direct medical costs as they directly influence resource allocation decisions by health insurers and healthcare institutions. Methods for calculating indirect costs (such as human capital or friction cost methods) exhibit significant heterogeneity, which could introduce uncertainty into the results. Given the substantial regional economic disparities in China, incorporating indirect costs might complicate the interpretation of study findings. Therefore, these costs are excluded from the base analysis, consistent with the aforementioned recommendation.

Policy Relevance: The economic evaluations used in China’s health insurance negotiations primarily focus on direct medical costs, ignoring productivity losses ensures alignment with policymakers' decision-making frameworks. Furthermore, the core objectives of the National Vital Basic Public Products Program (NVBP) are to reduce drug prices and medical expenditures rather than indirect social costs, further supporting the scope of cost accounting in this study.

7. There is no scenario analysis. Was it possible to no structural uncertainties around the model leading to no need for scenario analysis?

Response: We appreciate the reviewer's attention to this issue. Although the probabilistic sensitivity analysis (PSA) indicates that the model is robust for most parameters, we have supplemented our analysis with scenario analyses:

Reduction in Antiviral Treatment Duration:

Simulations were conducted for antiviral treatment durations of 40 years and 20 years. The results show that TAF (under NVBP) maintains cost-effectiveness in both scenarios.

Adjustment of Discount Rates:

The conclusions remain unchanged when discount rates are set at either 3% or 7%.

Variation in Monitoring Costs for TDF and TAF:

Even when monitoring costs for TDF and TAF fluctuate by ±50%, TAF (under NVBP) remains the optimal strategy.

Detailed results are provided in the supplementary material, Figure S1. We have supplemented the description of the aforementioned content in the results section. (Page 6. Line 253-257)

8. The details on one-way SA should be added. What were the ranges reported in Table 1? 95%CI, actual ranges? How did the authors handle inputs which had no range (or CI) reported?

Response: We appreciate the reviewer's attention to the details of the sensitivity analysis. We have added the following content in the "Methods - Sensitivity Analyses" section to clarify the methods for determining parameter ranges and strategies for handling parameters without specified ranges:

Principles for Determining Parameter Ranges: For data supported by literature (the majority of data), ranges for parameters such as health state transition probabilities (e.g., CHB→cirrhosis), utility values, and complication costs are directly derived from the reported 95% confidence intervals (95% CI) or actual extremes from the literature; Handling Missing Data: Parameters that completely lack range data (such as liver transplant rates for liver cancer patients) or fixed parameters determined by policy (such as drug prices under NVBP policy) are included in the cost-utility analysis (CUA) as deterministic parameters. However, to validate the robustness of the CUA results, we set the parameter range from 0 to the fixed parameter value during sensitivity analysis to explore potential impacts on the model. (Page3-4, lines 126-134)

One-way sensitivity analysis shows that key parameters (e.g., treatment initiation age, discount rate) contribute over 96.7% to the outcome (Figure 3), while fluctuations in other parameters have a minor effect (<2.16%) on net monetary benefit (NMB), indicating reasonable parameter range settings. A probabilistic sensitivity analysis (PSA) involving 100,000 Monte Carlo simulations verified the model's robustness against parameter uncertainty (Figures 4-6).

9. Supplementary 1 could be moved to main table or incorporate into Table 2.

Response: We thank the reviewer for this constructive suggestion. In response, we have integrated the data from Supplementary Table 1 (mortality reduction) into the main text's Table 2 by adding a "ΔDeath" column, thereby providing a more comprehensive and intuitive presentation of clinical outcome differences (Table 2).

10. Table 2 NVBP or NVBP??? Use it consistently throughout the manuscript.

Response: We appreciate your feedback. We have standardized the use of "NVBP" (New Volume-Based Procurement Policy, NVBP Policy) throughout the entire manuscript and have corrected the abbreviations in the tables and figures accordingly.

11. Fully incremental analyses assessing cost-effective among the NA(s) should be considered.

Response: We appreciate the reviewer's important suggestions. In response to your concerns regarding fully incremental analyses, we have supplemented detailed pairwise comparisons of strategies in the "Results" sections to comprehensively assess the cost-effectiveness hierarchy of different NAs treatment strategies. We have added table 3 and specific revisions as follows:

Fully incremental cost-effectiveness analysis:

Moreover, we conducted a comprehensive fully incremental cost-effectiveness analysis separately under the conditions without NVBP and with NVBP (Table 3). Which systematically compares the incremental cost-effectiveness ratios (ICERs) for all strategy combinations. The results show that the ICER of TAF(NVBP) compared to TDF(NVBP) is -1,352.86 USD/QALY, indicating that TAF(NVBP) is a dominant strategy with lower costs and higher health outcomes. The ICER of TAF(NVBP) compared to ETV(NVBP) is -369.12 USD/QALY, also showing dominance. The ICER of TDF(NVBP) compared to ETV(NVBP) is -254.35 USD/QALY, further confirming the ranking of strategies. In probabilistic sensitivity analysis (PSA), TAF(NVBP) was superior to TDF(NVBP) in 54.23% of simulations (Figure 5C) and superior to ETV(NVBP) in 86.24% of simulations (Figure 5B), establishing a clear cost-effectiveness hierarchy: TAF(NVBP) > TDF(NVBP) > ETV(NVBP).

Comparison with Baseline Strategy, at a willingness-to-pay (WTP) threshold of 37,500 USD/QALY, the ICER of TAF(NVBP) compared to non-NAs BSC is -745.62 USD/QALY, indicating both cost savings and health gains (Table 2). The fully incremental analysis further confirms the optimal position of TAF(NVBP) among all strategies; even when the WTP threshold is reduced to 12,500 USD/QALY, the probability of it being the optimal strategy remains above 51% (Figure 6). (Page 5-6. 204-220)

12. CEA frontier might help authors to better report the findings.

Response: We appreciate the reviewer's suggestions. We have updated Figure 2 to a Cost-Effectiveness Frontier plot to more intuitively display the relationship between economic efficiency and health outcomes for each strategy. Specific revisions are as follows:

New and Adjusted Figures:

In Figure 2, the x-axis represents cumulative QALYs, and the y-axis represents cumulative costs. All strategies (TAF, TDF, ETV, with/without NVBP) are plotted relative to Non-NAs BSC. Three dashed lines through the BSC strategy point represent the range of Chinese willingness-to-pay (WTP) thresholds (12,500-37,500 USD/QALY). The solid line connecting dominant strategies forms the efficiency frontier, with the theoretical boundary at 37,500 USD/QALY shown by the solid line extending to the right from TAF(NVBP). This clearly shows that TAF(NVBP) is positioned at the farthest bottom-right on the frontier, indicating it has the highest health outcome and the lowest cost (Figure 2).

Revisions in Results Section:

---

## [Decision Letter · Decision Letter 1]

Tenofovir alafenamide is Superior to Tenofovir disoproxil fumarate and Entecavir in Cost-Effectiveness of Treatment of Chronic Hepatitis B in China With New Volume-Based Procurement Policy

PONE-D-24-56441R1

Dear Dr. Lu,

We’re pleased to inform you that your manuscript has been judged scientifically suitable for publication and will be formally accepted for publication once it meets all outstanding technical requirements.

Kind regards,

Yury E Khudyakov, PhD

Academic Editor

PLOS ONE

Additional Editor Comments (optional):

Reviewers' comments:

Reviewer's Responses to Questions

**Comments to the Author**

1. If the authors have adequately addressed your comments raised in a previous round of review and you feel that this manuscript is now acceptable for publication, you may indicate that here to bypass the “Comments to the Author” section, enter your conflict of interest statement in the “Confidential to Editor” section, and submit your "Accept" recommendation.

Reviewer #1: All comments have been addressed

2. Is the manuscript technically sound, and do the data support the conclusions?

Reviewer #1: Yes

3. Has the statistical analysis been performed appropriately and rigorously? 

Reviewer #1: Yes

4. Have the authors made all data underlying the findings in their manuscript fully available?

Reviewer #1: Yes

5. Is the manuscript presented in an intelligible fashion and written in standard English?

Reviewer #1: Yes

6. Review Comments to the Author

Reviewer #1: The authors responded my comments well. It is acceptable for publication in this revised version of the manuscript.

7. PLOS authors have the option to publish the peer review history of their article (what does this mean? ). If published, this will include your full peer review and any attached files.

**Do you want your identity to be public for this peer review?** For information about this choice, including consent withdrawal, please see our Privacy Policy .

Reviewer #1: **Yes: ** Piyameth Dilokthornsakul

---

## [Editor Report · Acceptance letter]

PONE-D-24-56441R1

PLOS ONE

Dear Dr. Lu,

I'm pleased to inform you that your manuscript has been deemed suitable for publication in PLOS ONE. Congratulations! Your manuscript is now being handed over to our production team.

Kind regards,

on behalf of

Dr. Yury E Khudyakov

Academic Editor

PLOS ONE